# Cryo-EM structures reveal two distinct conformational states in a picornavirus cell entry intermediate

**Pranav N. M. Shah**[1¤a], **David J. Filman**[1], **Krishanthi S. Karunatilaka**[1], **Emma L. Hesketh**[2], **Elisabetta Groppelli**[3¤b], **Mike Strauss**[4], **James M. Hogle**[1]*

**1** Department of Biological Chemistry and Molecular Pharmacology, Harvard Medical School, Boston, MA, United States of America, **2** Astbury Centre for Structural Molecular Biology, School of Molecular & Cellular Biology, Faculty of Biological Sciences, University of Leeds, Leeds, United Kingdom, **3** School of Molecular and Cellular Biology, Faculty of Biological Sciences, University of Leeds, Leeds, United Kingdom, **4** Department of Anatomy and Cell Biology, McGill University, Montreal, Canada

¤a Current address: The Division of Structural Biology, The Henry Wellcome Building for Genomic Medicine, Roosevelt Drive, Oxford, United Kingdom
¤b Current address: Institute of Infection and Immunity, St George's, University of London, Cranmer Terrace, Tooting, London United Kingdom
* james_hogle@hms.harvard.edu

**Data Availability Statement:** Cryo-EM maps and masks for m135, r135, h135, r135+m25 and r135 +m37 conditions are deposited in the EMDB with accession codes EMD- 20275, EMD-20276, EMD-

## Abstract

The virions of enteroviruses such as poliovirus undergo a global conformational change after binding to the cellular receptor, characterized by a 4% expansion, and by the opening of holes at the two and quasi-three-fold symmetry axes of the capsid. The resultant particle is called a 135S particle or A-particle and is thought to be on the pathway to a productive infection. Previously published studies have concluded that the membrane-interactive peptides, namely VP4 and the N-terminus of VP1, are irreversibly externalized in the 135S particle. However, using established protocols to produce the 135S particle, and single particle cryo-electron microscopy methods, we have identified at least two unique states that we call the early and late 135S particle. Surprisingly, only in the "late" 135S particles have detectable levels of the VP1 N-terminus been trapped outside the capsid. Moreover, we observe a distinct density inside the capsid that can be accounted for by VP4 that remains associated with the genome. Taken together our results conclusively demonstrate that the 135S particle is not a unique conformation, but rather a family of conformations that could exist simultaneously.

## Author summary

Nonenveloped viruses need to provide mechanisms that allow their genomes to be delivered across membranes. This process remains poorly understood. For enteroviruses such as poliovirus, genome delivery involves a program of conformational changes that include expansion of the particle and externalization of two normally internal peptides, VP4 and the VP1 N-terminus, which then insert into the cell membrane, triggering endocytosis and the creation of pores that facilitate the transfer of the viral RNA genome across the

20469, EMD-20546 and EMD-20474 respectively. Atomic co-ordinates for all the maps except r135 +m37 can be accessed from the PDB with the following codes 6P9O (m135), 6P9W (r135), 6PSZ (h135), 6Q0B (r135+m25).

**Funding:** J.M.H was funded by National Institutes of Health grant AI020566 (https://www.nih.gov/). The funders had no role in study design, data collection and analysis, decision to publish, or preparation of the manuscript.

**Competing interests:** The authors have declared that no competing interests exist.

endosomal membrane. This manuscript describes five high-resolution cryo-EM structures of altered poliovirus particles that represent a number of intermediates along this pathway. The structures reveal several surprising findings, including the discovery of a new intermediate that is expanded, but has not yet externalized the membrane interactive peptides; the clear identification of a unique exit site for the VP1 N-terminus; the demonstration that the externalized VP1 N-terminus partitions between two different sites in a temperature-dependent fashion; direct visualization of an amphipathic helix at the N-terminus of VP1 in an ideal position for interaction with cellular membranes; and the observation that a significant portion of VP4 remains inside the particle and accounts for a density feature that had previously been ascribed to part of the viral RNA. These findings represent significant additions to our understanding of the cell entry process of an important class of human pathogens.

## Introduction

Poliovirus is a small (~30nm) non-enveloped, positive sense, single-stranded RNA virus. It belongs to the genus Enterovirus of the Picornaviridae, and is the causative agent of poliomyelitis [1]. The icosahedral capsid of the virus, which is made of four proteins, namely VP1, VP2, VP3, and VP4, encloses the viral genome [2]. The capsid is structurally similar to other members of the family such as Rhinovirus, Coxsackievirus A16, Enterovirus 71 and Enterovirus D68, the latter three having caused recent epidemics of hand-foot-and-mouth disease, and even flaccid paralysis in China and the United States [1,3,4]. Therefore, studies on poliovirus are not only crucial to support the global effort to eradicate poliovirus, but also to tackle important emerging pathogens.

Poliovirus entry into cells is initiated when the virus interacts with its receptor, CD155, also known as poliovirus receptor (PVR) [5], an immunoglobulin-like molecule that is expressed at the intercellular junctions of epithelial cells [6]. Receptor engagement is concomitant with the loss of the "pocket factor" [7], a stabilizing ligand [8] that binds in the hydrophobic core of VP1 [9]. Structurally, receptor binding results in an icosahedral expansion of the capsid by 4% [10], yielding a 135S, or A- particle [11–16]. These particles are metastable, infectious [17,18] and RNA-containing. Therefore, it is thought that the 135S particle represents a critical step along the pathway to infection [18]. Unlike mature native virions (160S particles), whose capsids form a closed surface, the expanded 135S particles have holes in the capsid at the icosahedral 2-fold and quasi-3-fold axes. These openings permit the externalization of the membrane-interactive N-terminal extension of capsid protein VP1 [11,14] in poliovirus and in the 135S particles of other enteroviruses [19–26] and presumably VP4, which is myristoylated [27]. *In vitro* studies with model membranes have shown that these polypeptide chains are responsible for anchoring the virus to the host cell membrane [11,28]. Earlier tomographic studies from our group demonstrated umbilical connections between the virus and receptor-decorated liposomal membranes [29] that were wide enough to accommodate single-stranded genomic RNA. This could be the manner in which the genome is protected from the degradative effects of RNases during RNA transfer [30]. However, the molecular and structural underpinnings of this process still remain elusive.

In the present study we report five high-resolution reconstructions of the 135S uncoating intermediate particles, with refined atomic models for four of them.

Taken together, our results describe a multi-step model for picornavirus capsid uncoating and subsequent steps in genome translocation.

## Results

### Three structures produced by incubation with mAb

Initially we set out to investigate the structure of the "breathing" poliovirus particle. Previous studies showed that native poliovirions, when incubated at 37˚C (but not at 25˚C), transiently expose membrane-interactive polypeptides VP4 and the N-terminus of VP1 [31]. These two exposed polypeptides could be trapped outside the capsid by specific antibodies, some of which neutralized infectivity [31]. The release of the antibodies by freeze-thaw restored infectivity, demonstrating that the process was reversible [31]. An early attempt to characterize the breathing particle, trapped using a Fab that recognized residues 39–55 of VP1 [32], produced a low-resolution structure [33]. This structure revealed an RNA-containing virus particle whose capsid was trapped in an icosahedrally symmetric expanded state that was purported to be different from previously reported expanded forms of poliovirus. Weak and poorly ordered density located above the icosahedral 2-fold hole was attributed to peptide-bound Fab molecules getting "stuck" in random orientations when the VP1 N-terminus was partially withdrawn into the particle interior [33].

In the current study we have leveraged improvements in EM and computer hardware, data collection software, and image processing pipelines to re-examine with higher detail the antibody-trapped states relevant to the uncoating process, starting with the poliovirus "breathing" particle. This particle was produced by incubating native poliovirus with a commercially available monoclonal antibody (kind gift from Quidel Corp.) that was raised against a peptide corresponding to VP1 residues 42–55 (PALTAVETGATNPL). Although the antibodies used in this and previous studies recognize the same region of the N-terminus of VP1, this antibody differs from the one previously used, which is no longer available. Thus, in contrast to the earlier antibody which showed significant binding when incubating virus with the Fab, we could only obtain significant decoration of particles using the intact monoclonal antibody (mAb) provided by Quidel. Initial screening of conditions to trap the N-terminus outside the virus using the mAb at 37˚C yielded very few decorated particles. Therefore, we performed additional cryo-EM screenings of the virus-mAb complexes for durations of up to 120 mins. at temperatures of 38˚C and 39˚C. An optimal number of complexes was observed after a 90-minute incubation of the virus with the mAb at 39˚C.

Under those conditions, three main structural classes were obtained (Fig 1A), each containing roughly one-third of the usable particle images. Class 1 was not labeled by antibodies and was indistinguishable from previously solved 160S native virus [2], having an RNA core; VP4 and the N-terminus of VP1 bound on the inner surface of the capsid; and a closed protein shell that lacked large holes (left column, Fig 1A). Classes 2 and 3 both consisted of capsids that were expanded by about 4%, with large holes at the 2-fold axes. The class 2 particles appeared to contain a full complement of RNA (middle column, Fig 1A), whereas class 3 appeared similar to previously-studied 80S empty capsids [10,34,35] (rightmost column, Fig 1A), which lack RNA, VP4, order in residues 1–68 of VP1, and which have either unobstructed or partially obstructed holes at the quasi-3-fold axes.

**The 80S-like class.**   In the icosahedral reconstructions of the class 3 particles, the density for the Fab portion of the bound antibody (for simplicity we will refer to this here and in subsequent sections as the Fab) was a flat roundish feature whose connection with the capsid was not directly apparent (rightmost column, Fig 1A). This density enclosed only the lower half of the Fab and was only visible at a much lower contour level than the capsid protein, suggesting both low occupancy, and variability in the position and angle of the Fab with respect to the virus. The position of the Fab, over the 2-fold hole, was similar to one of two sites for Fab binding in the low-resolution structure of an 80S-Fab complex described previously [36].

A.

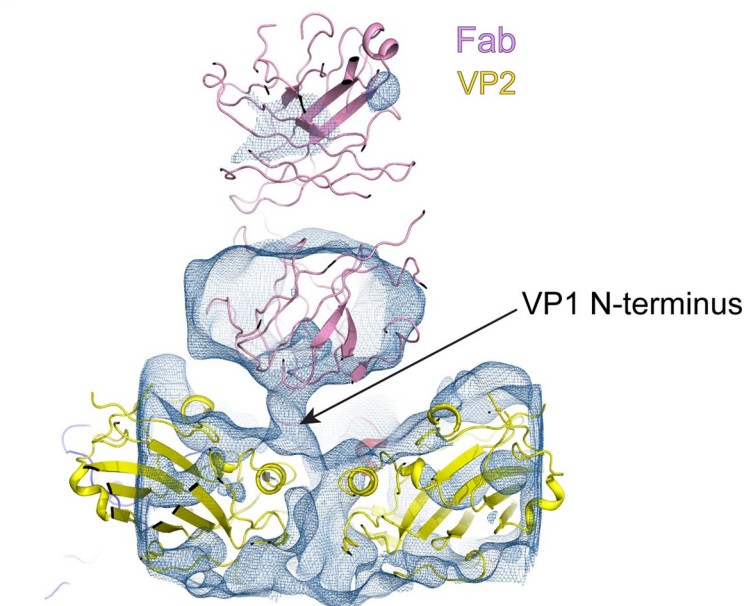

B.

**Fig 1. Anti-VP1 antibody trapped expanded particle.** (A) Top row: The 3D classes from the dataset, with undecorated 160S particles (left, blue border), the anti-VP1 mAb decorated 135S-like particle (middle, orange border) and the mAb decorated empty particle (right, yellow border). Second row: Radially colored isosurface representations of the refined particles are shown at a high contour, where the holes at the 2-fold are clearly observed. Third row, isosurface representations of the mAb-decorated 135S (center) and 80S particles (right) contoured at a lower level showing the diffuse density attributable to the bottom half of the Fab part of the mAb. Bar indicates the radial range that each color corresponds to. (B) Density for an asymmetric focus class calculated from the mAb decorated 80S particle is depicted in the context of the 2-fold related VP2 subunits (yellow) and a model of a Fab (pink). The tube-like density for the N-terminus of VP1 (arrow) is seen above one of the two 2-fold related VP2 helices.

Focused classification methods, recently introduced into cryoEM, often succeed in avoiding the averaging of symmetry-related areas that have significantly different density patterns [37–39]. To further assess the interaction of the Fab with the capsid, we performed an asymmetric focused classification on 542,460 asymmetric units of the virus, using a cylindrical mask that enclosed the 2-fold axis of the capsid as well as the Fab. While the capsid was resolved to 4.2 Å in the icosahedrally averaged maps, the focused class in (Fig 1B) resolved to 6.5 Å. This class had a distinct tube-like density extending from above the VP2 helix (residues 90–100) at the 2-fold axis to the underside of the Fab (Fig 1B). We interpret this density to be the N-terminus of VP1.

**135S-like particles (m135).**   Class 2 (which we will call m135) was the most interesting of the 3 classes (Fig 2 and middle column of Fig 1A), as it shared several of the physical characteristics of the canonical 135S particles. The m135 reconstruction was resolved to 2.8 Å resolution in the capsid region, after masking and "sharpening" (see Methods) (S1 Table, S1 Fig), and a detailed atomic model was constructed and refined for ordered amino acids in capsid proteins VP1, VP2, and VP3. Like a canonical 135S particle, the m135 reconstruction includes a significant central density for the RNA (but whose apparent icosahedral symmetry is an artefact of the calculation), lacks VP4 on the inner surface of the capsid, and has an expanded capsid with holes at the 2-fold and quasi-3-fold axes (middle column Fig 1A). In this structure and in all of the remaining 135S structures described below, the pocket factor is missing from the hydrophobic pocket of VP1 [9]. Similar to 135S-like particles for other enteroviruses [19–26], this structure has an additional portion of the N-terminal extension of VP1 (residues 64–69) that is ordered, and extends outward along the quasi-3-fold axis (Fig 2B–2D).

The density for the Fab is diffuse and rod-shaped, stretching between two 2-fold-related copies of the quasi-3-fold hole (Fig 2A and middle column of Fig 1A). The rod-like density feature is only evident above the noise level for the lower half of the Fab, which suggests variability in the position and orientation of the Fab. The weakness of the Fab density is not unexpected, given that the VP1 N-terminus is flexible, and entirely disordered in most non-native poliovirus structures. The weakness of the ordered Fab density (in the icosahedrally averaged maps) is consistent with a model in which specific Fab binding (to residues 42–55 of the VP1 N-terminus) at some of the quasi-3-fold sites near the virus surface is precluded by steric conflicts with previously bound Fabs.

In the high-resolution m135 structure, we see two general categories of structural change relative to native virus. One category corresponds to movement of the beta barrels of the major capsid proteins, VP1-3, away from one another when the capsid expands. As a consequence, several polypeptide segments (which, in native 160S virus, stabilize the "closed" capsid arrangement) detach from their neighbors and become disordered. On the inner surface of the capsid, these segments include VP4 and portions of the N-terminal extensions of VP1 and VP2. On the outer surface of the capsid, these segments include all C-termini of VP1, VP2, and VP3, as well as loops extending from the beta-barrel cores of VP1 and VP2 (see below). The general pattern of beta barrel shifts and polypeptide chain disordering that occurs upon the expansion of poliovirus has previously been analyzed [10,14,34,35], and is strikingly similar

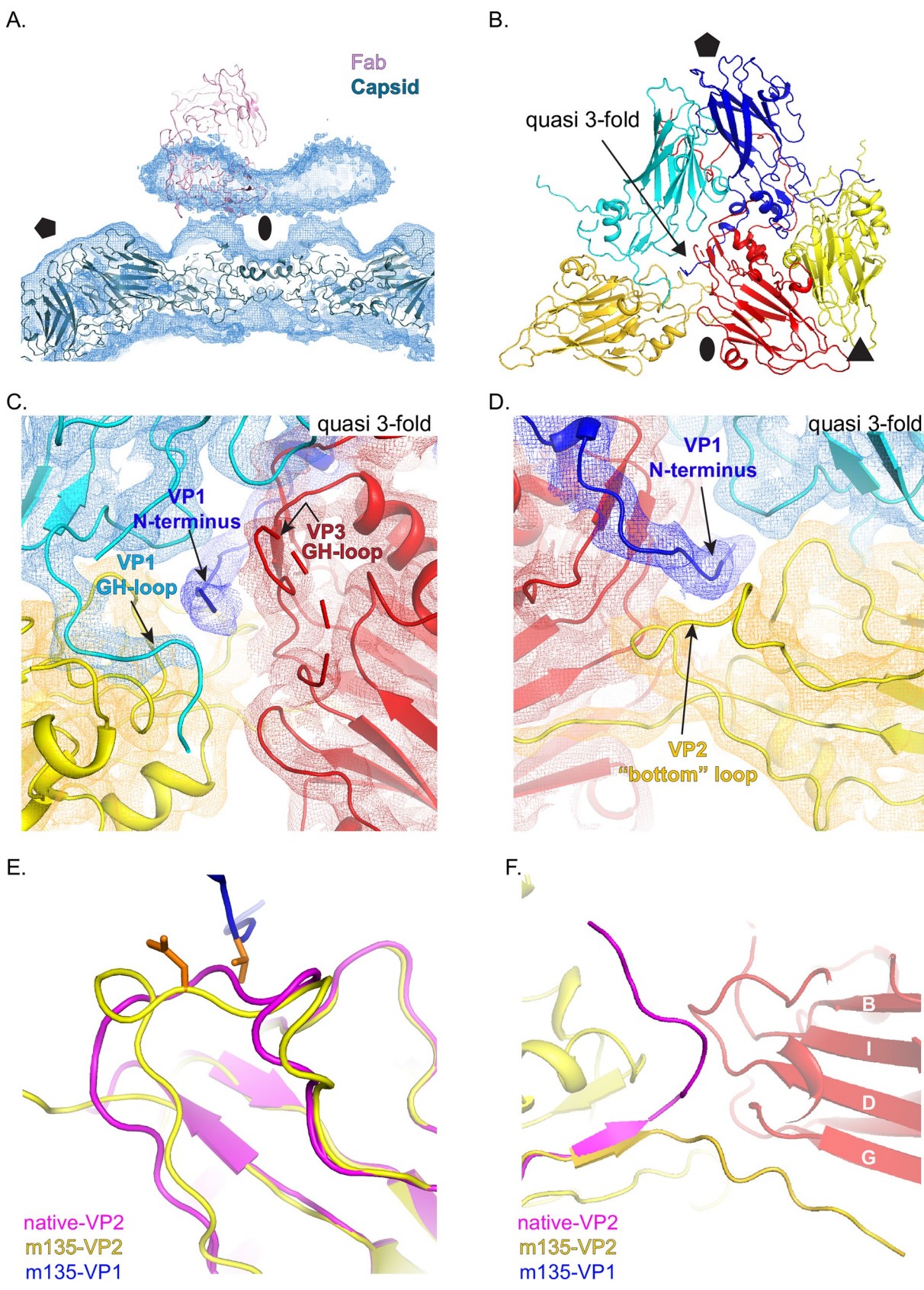

**Fig 2. Anti-VP1 antibody trapped genome containing particle (m135) is similar to the canonical 135S particle.** (A) Cage density depiction (σ = 0.25) of the mAb-decorated m135 structure in the vicinity of the 2-fold axes showing the rod shaped density for the lower half of the Fab portion of the mAb and the better ordered density for the capsid proteins of the virus (B) The canonical poliovirus protomer consisting of VP1 (blue), VP2 (yellow) and VP3 (red) is shown juxtaposed with 5-fold related copies of VP1(cyan) and VP2 (gold). The cyan, gold, and red copies together form the "5-3-3 triangle" having one 5-fold and two 3-fold symmetry axes at its corners. The quasi 3-fold lies at the center of this triangle. Panels (B) and (C) are viewed from outside the virus. (C) The N-terminus of VP1 (dark blue) is seen exiting the virus capsid through the quasi 3-fold hole. To accommodate this, the GH loop of VP3 (labeled) becomes rearranged and partially disordered. Panels (D-F) are viewed from inside the virus. (D) The exiting N-terminus (dark blue) contacts the tip of the VP2 bottom loop (labeled) (E) The bottom loop of VP2, in m135 (yellow), is rearranged relative to native VP2 (magenta) (F) In m135, the C-terminus of VP2 (dark yellow) becomes ordered on the capsid inner surface, rather than exiting at the 2-fold, as in native virus (magenta).

to the shifts seen in the structures of 135S particles of other enteroviruses [19–26]. A specific listing of ordered and disordered residues is tabulated (see S2 Table) for the m135 atomic model, and for the three other atomic models described below.

The second, particularly interesting, set of structural changes involves the polypeptide segments that contact the VP1 N-terminus as it exits through the quasi-3-fold hole. Notably, the GH loop of VP3 (residues 176–185) uncoils, changing from a compact structure that plugs the quasi-3-fold in virions into a more extended structure that leaves room for residues 62–68 of VP1 (residues 1–61 were not modellable in this density) to thread through the quasi-3-fold (Fig 2C and 2D). Additionally, the "bottom" loop of VP2 (residues 26–58), which encircles the bottom of the VP2 beta barrel, rearranges at its tip (residues 43–55) to move out of the way of the externalizing VP1 N-terminus (Fig 2D and 2E). Finally, a portion of the GH loop of VP1 (residues 216–236) rearranges to facilitate the externalization of the VP1 N-terminus; and the C-terminal extension of VP2 (residues 262–272) retracts from its position on the outside of the virion to bind to the inner surface of the capsid (Fig 2F). The movement of the VP1-3 beta barrels is strikingly consistent with all of the 135S structures reported here and in those previously published for poliovirus and other enteroviruses [14,18–20,23–26,40] (see ordered residue lists, S2 Table). In particular, the four key contact areas (for the VP1 N-terminus) (Fig 2C–2F) either change their conformations or become disordered as a result of the 160S-to-135S conversion.

Finally, we considered the possibility that our antibody-trapped "breathing" particles would regain infectivity if the antibody were released by freeze-thawing, as was reported in the original report of "breathing" of poliovirus [31]. To test this possibility, virus samples were incubated with VP1 antibodies either at 39°C (where breathing is expected) or 25°C (where breathing is not expected) for 1.5 h (the same as for the EM experiment) and immunoprecipitated with Protein A/G coated magnetic beads at room temperature. Unbound virus was washed away, and the bound virus was released from the beads by freeze-thawing the solution. Next, the released virus was serially diluted and used to infect naïve Vero cells. Greater cytopathic effect was observed in virus particles retrieved from incubating the virus with the antibody at higher temperature than the virus particles retrieved from the room temperature incubation (S2A and S2B Fig). This suggests that some of the antibody-bound particles must have retained a full or nearly full complement of proteins (presumably including VP4) and remained infectious, even when accounting for non-specific binding of native virus to the antibody.

## Comparison of m135 particles with "canonical" 135S particles

Historically, three methods were used to convert 160S virus to 135S particles, and the resulting particles were not distinguishable by any of the tests then available. One method involved the binding of virus by the poliovirus receptor at the cell surface at 37°C in a natural infection [11–13]; the second involved incubation of virus with the soluble ectodomain of the receptor

at 37˚C [41,42]; and the third involved heating 160S particles at 50˚C in the presence of divalent cations that stabilize the meta-stable 135S intermediate [17,43]. To assess whether or not the m135 particle was indeed 135S-like, we decided to solve cryo-EM structures for both the soluble-receptor-triggered and heat-triggered 135S particles. Consistent with our "standard recipes" for producing 135S particles [8,17], we incubated native virus for 3 min, either with solubilized PVR at 37˚C, or alone at 50˚C. Note that PVR falls off the virus once expansion has been triggered. Receptor-triggered and heat-triggered 135S structures (designated "r135" and "h135", respectively) were solved at 3.2 Å resolution (after density-based masking and sharpening) (S1 Table, S1B and S2C Fig).

As expected, the capsids of both r135 and h135 were expanded by 4%, with holes at the 2-fold axes, were full of RNA, and failed to show well-ordered native-like VP4 pentamers lying flat on the inner surface of the capsid (Fig 3A). The r135 and h135 structures were qualitatively similar. There are no instances where several residues in succession are ordered in one model and disordered or differently ordered in the other. Minor differences between the two models include the occasional shift, by one or two residues along the polypeptide chain, where an order-to-disorder transition occurred, and scattered (rather than systematic) differences in the sets of "buildable" residues in native-like or non-native-like conformations (which are listed in S2 Table).

Surprisingly, both r135 and h135 maps lacked a visible extension of the VP1 N-terminus through the quasi-3-fold hole, which was supposed to be the structural hallmark of these particles [11,14,40]. In both cases, residue 69 was the first ordered residue of VP1; residues 43–53 of the bottom loop of VP2 (which supports the VP1 N-terminus) were disordered; and the quasi-3-fold hole was partially blocked by the GH loop of VP3 (Fig 3B). The lower portion of the loop was similar to that seen in native 160S virus (Fig 3C), but the upper portion was different from either the 160S structure or previously described 135S structures in which the VP1 N-termini were exposed. In short, while the literature [20,22,24,26] (and our own experience [14]) had led us to expect a 3 min incubation to produce an expanded particle with the VP1 N-termini extended, the 3 min incubation in this study produced an expanded and fenestrated capsid in which most or all of the VP1 N-termini remained inside the capsid. Therefore, we propose that the category of poliovirus particles that was previously called "135S" most likely consists of distinct states, namely, an "early state", wherein the capsid has undergone an expansion but most copies of the VP1 N-terminus are on the inside of the capsid and the quasi-3-fold hole still remains blocked, and a subsequent "late state", characterized by the opening of the quasi 3-fold hole and the emergence of many copies of the VP1 N-terminus through them. In support of this possibility, a careful re-reading of the literature [17,31,40] showed that 135S particles with extended N-termini were typically detected only after extensive purification and/or prolonged incubations. Interestingly, an expanded particle in which the VP1 N-terminus remained inside the capsid has also been observed as a minority subpopulation (~5% of the particles) in preparations of 135S particles of enterovirus D68 [19]. However, in this subpopulation of the D68 135S particles the entire VP1 N-terminal extension is well ordered suggesting that the "early" state of the 135S-like particles may itself be comprised of two steps. Thus, in a first step the particle undergoes expansion with the VP1 N-terminus remaining in a well-ordered native-like arrangement on the inner surface of the expanded capsid; this is followed by a second step in which the VP1 N-terminal extension dissociates from the inner surface, prior to being externalized.

## Receptor-induced 135S particles incubated with mAb at 25˚C

To test our hypothesis, that VP1 N-termini (from expanded particles) become externalized after an extended time, we followed the method described in Tsang et al [44], first incubating

A.

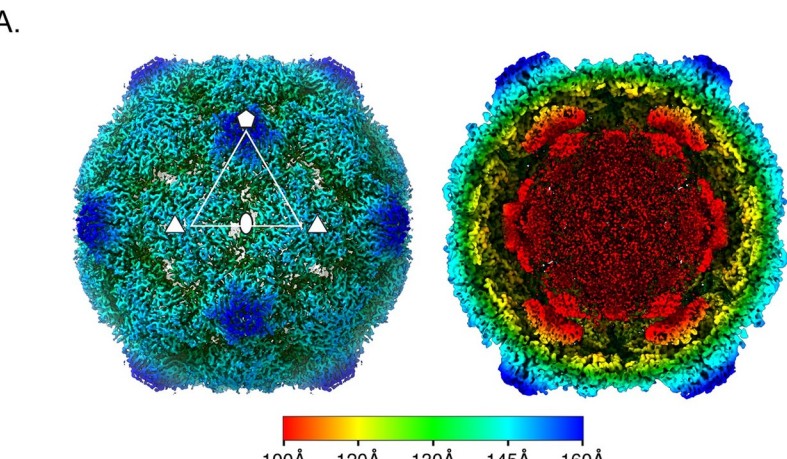

B.

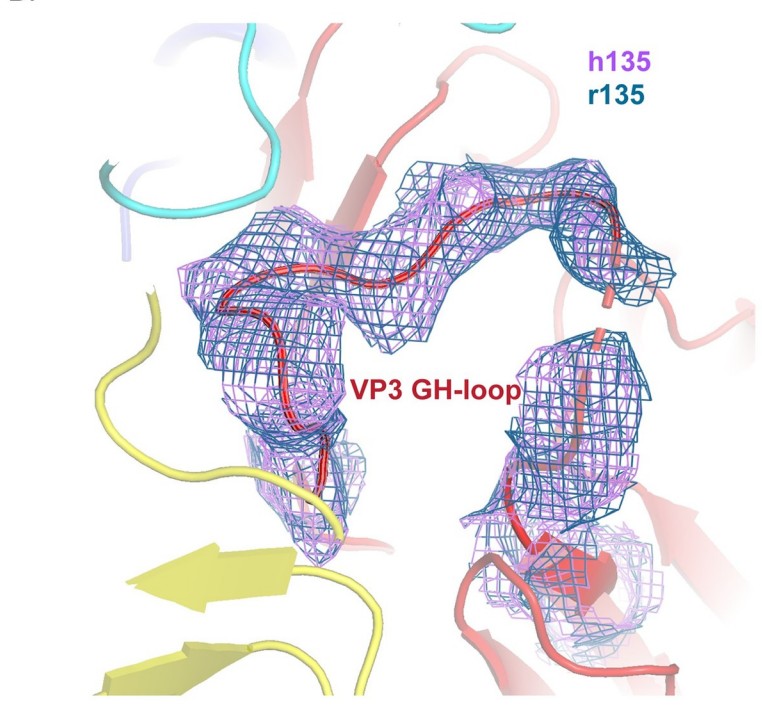

C.

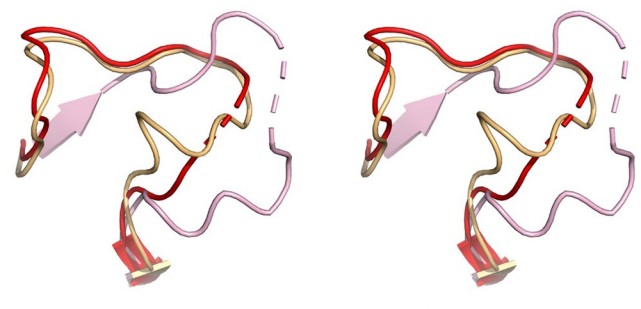

**Fig 3. Receptor-catalyzed, and heat-triggered 135S particles (r/h135).** (A) Radially colored isosurface rendering of the "early" 135S particle at a high contour ($\sigma = 5$) (left panel) reveals an opening at the 2-fold axis. The panel on the right is shown at a lower contour level ($\sigma = 2$) with a clipping plane inserted to highlight the re-organised density inside the capsid. These two panels are based on the r135 map, because the two structures are virtually identical. Bar indicates the radial range that each color corresponds to. (B) Close up of the quasi-3-fold region. Density for the GH loop of VP3, overlaid on the model, is very similar in the r135 (blue) and h135 (purple) maps. In this conformation, the GH loop of VP3 plugs the quasi-3-fold hole. (C) Stereo views compare the GH loops of VP3 from the r135 (red), m135 (salmon), and native (orange) structures. The r135 conformation is more similar to the native conformation, as opposed to the m135 model where the GH loop becomes more extended and forms new beta-sheet interactions with a radially-oriented segment of the VP3 D strand.

the virus with receptor for 3 mins at 37°C, followed by a 1hr incubation with antibody at 25°C. This protocol was carried out twice, once with a smaller "screening" data set, and once with a larger one. Data collection parameters are summarized in S1 Table.

As predicted, incubation of the pre-formed "early 135S particle" with antibody for 1hr at 25°C did decorate the virus particles with Fab-shaped density features. (Fig 4A). Surprisingly, the Fab portions of the antibody molecules did not bind at quasi-3-fold positions, to produce a low-level rod-shaped density, such as that seen in the m135 map (Fig 2A and Fig 1A middle column). Instead, the Fab density in the icosahedrally-averaged map was poised above the 2-fold hole. Its position was similar to that of the much weaker hemispherical density in the decorated 80S particles (rightmost column, Fig 1A). However, in the new "r135+m25" reconstruction, density for the Fab was much better ordered, with the upper half of the Fab domain visible, and separated from the lower half by an appropriately placed gap in the density between subdomains (Fig 4B). Unlike the decorated 80S map, where the Fab may have been freely spinning around the viral 2-fold axis, here the Fab appears to have a preferred orientation, with each of its two hinges sitting on a (partly disordered) EF loop of VP2. (Note that the Fab envelope is a sufficiently low-resolution feature that imposing an icosahedral twofold operation on the Fab density envelope had only a minimal effect on the density shape). The density for the Fab being weaker than that of the capsid can be attributed, at least in part, to the fact that at most 30 copies of the Fab can bind to the virion over the 2-fold axis. Both the contour level needed to visualize the Fab and the quality of fit of this density to the Fab model indicated higher occupancy and more structural consistency in this complex than we saw in our other complexes with mAb.

In the r135+m25 icosahedrally averaged map, and in all of the focused classes that were calculated from the r135+m25 image stack, we looked under the bottom surface of the Fab, trying to find density for a peptide connection (perhaps similar to the density connections that we saw in the focus classes of antibody-decorated 80S particles (Fig 1B). Instead, the bottom surface of the Fab (above the 2-fold axis) consistently lacked an obvious connection; and the 2-fold hole was occupied by a density that we tentatively identify as the partially retracted C-terminus of VP2 (Fig 4B). In contrast, in focused classifications centered on the 2-fold axis, including the quasi-3-fold holes, we observed 61% of the asymmetric particles exhibiting a clear density for the N-terminus of VP1 emerging from the quasi 3-fold hole (Fig 4C), with the GH loop of VP3 in an extended conformation as described in the m135 model (Fig 2C).

**Visualizing the exposed N-terminal helix of VP1.** Interestingly, we observed a focused class for the screening data comprising of 21% of the 966,840 asymmetric units of the virus where the lower half of the Fab contacts both VP2 shoulders, but asymmetrically tilts towards one of the shoulders (Fig 4D). This causes the top half of the Fab to become less well ordered compared to the lower half of the molecule. In this class, the tilted Fab leans against a long, straight, upward-projecting density feature. The projecting density is unusual in its appearance with one wide dimension and a flat dimension (Fig 4D).

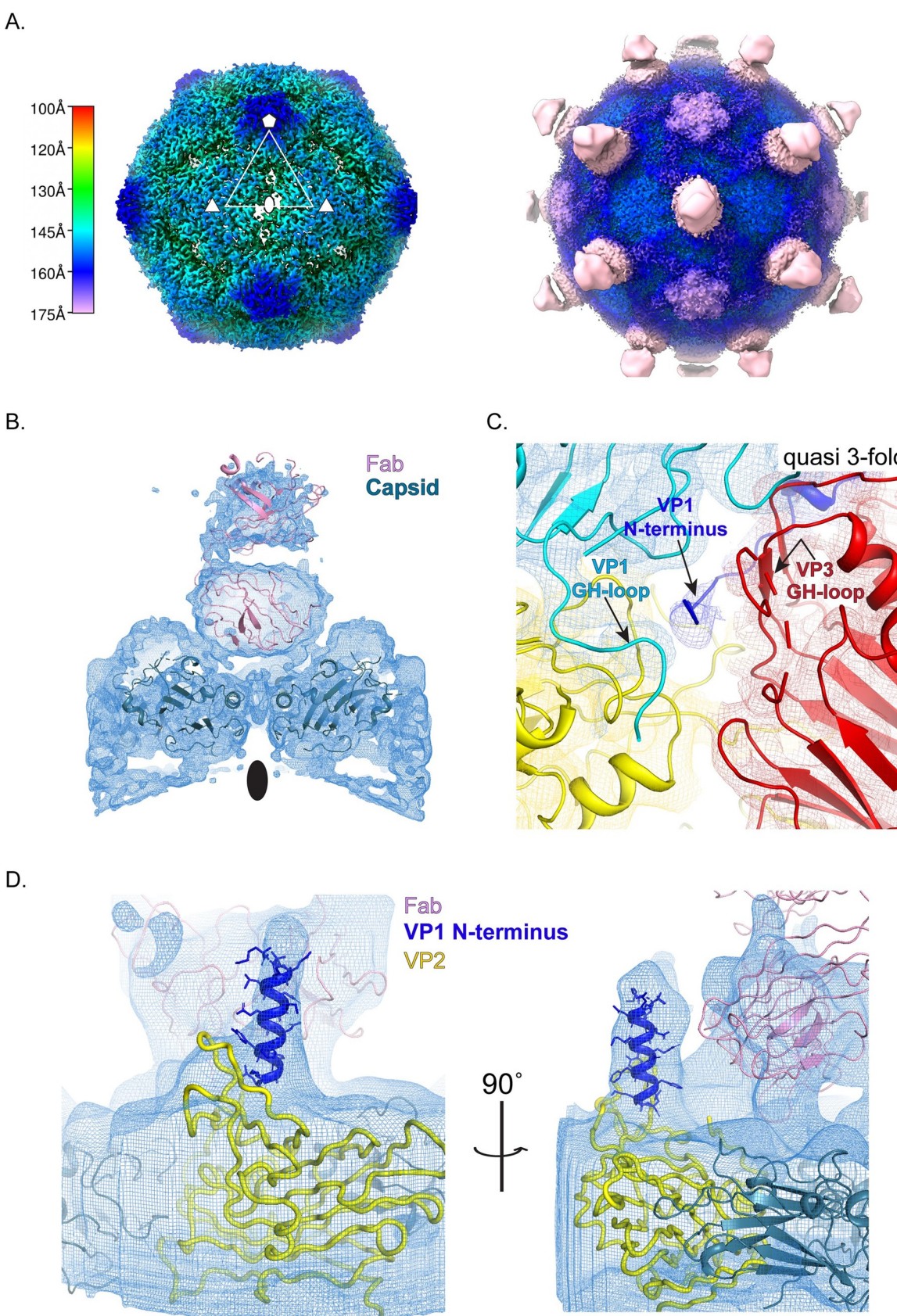

**Fig 4. Receptor catalyzed 135S particle incubated with the anti-VP1 antibody at 25°C (r135+m25).** (A) Radially colored isosurface rendering of the "late" 135S particle shown at high contour (σ = 5) (left) to highlight the hole at the 2-fold axis and at a lower contour (σ = 0.5) (right) to highlight binding of the Fab (pink) relative to the capsid. Bar indicates the radial range that each color corresponds to. (B) The interaction between the Fab density and the capsid is better resolved following a focused classification centered around the 2-fold symmetry axis of the capsid. (C) A focus map from (B) contoured at a σ = 3 compared with the m135 model, helps to visualize the N-terminus of VP1 (blue) exiting the capsid from the quasi-3-fold hole. The density in this region is poorly resolved in the icosahedral averaged reconstruction (D) A focused class obtained from the screening data contoured at σ = 3, using a cylindrical mask centered on the 2-fold axis. This map clearly shows a column of density that accommodates the first 20 amino acids of the N-terminus of VP1 (blue). The helix bound to the top surface of VP2 (yellow) is stabilized in a unique orientation by contacts with the Fab (pink). Clearly, the portions of VP2 that bind the base of the helix must be rearranged, in some unknown way that differs from the native conformation of VP2 that is shown (yellow).

The observed shape of the density closely matches the shape of an idealized alpha helix built with the sequence of the first 20 residues of VP1 (the amphipathic character of this helix is known to be evolutionarily conserved [11]). The helical density observed in our map (Fig 4D) appears on the top surface of the VP2 EF loop, in the same location that a difference density peak was previously observed between low-resolution structures of V8 protease-treated and untreated 135S particles [40] (S3B Fig). Similar column-like density features were seen in some of our other focus classes, but the density was never quite as convincing as it was in this example.

Taken together these data suggest that the Fab in r135+m25 particles is bound to a copy of the VP1 N-terminus that has exited at a nearby quasi-3-fold axis, and that its location above the 2-fold hole is determined by interactions of the framework region of the Fab with the extreme N-terminal residues of VP1 and the VP2 GH loops.

## Receptor-induced 135S particles incubated with mAb at 37°C

To account for the difference in the binding pattern of the Fab between the m135 and r135 +m25 reconstructions, we postulated that lowering the temperature had an effect on the pattern of Fab binding. To that end the experiment was performed in the same manner as the r135+m25 conditions, except that the temperature for incubating with antibody was maintained at 37°C. The Fab density that was observed (Fig 5A and 5B) appeared to be a superposition of the m135 and r135+m25 binding patterns, Thus, the density for the Fab is rod-like (as it was in the m135 reconstruction), but there is also significant density above the 2-fold axis which includes density corresponding to the upper half of the Fab (as was observed in the r135 +m25 reconstruction). The reduced occupancy of well-ordered Fab molecules at the 2-fold site (in r135+m37, relative to r135+m25) suggests that the higher temperature of incubation with antibody destabilizes the interaction of Fab with the VP2 GH loop. As was the case in the m135 structure, the r135+m37 reconstruction has readily interpretable density for the VP1 N-terminus exiting at the quasi-3-fold hole. Given that the r135+m37 map appears to be a lower-resolution (see S1 Table) superposition of two better-determined structures, and not relevant to biological infection, we elect not to report an atomic model for it.

## Possible pentamers of capsid protein VP4

The small myristoylated capsid protein VP4 [27], is believed to play an important role in the protection of viral RNA (as RNA crosses the gap between virus and membrane) and in the penetration of host cell membranes [45–47]. In 160S virions, prior to capsid expansion, pentamers of poliovirus VP4 lie flat on the inner surface of the capsid, cupping the VP3 beta tube in a ring of five myristates [2,9,27] (Fig 6, leftmost panel). The N-terminal residues (1–45) of VP4 intertwine in a pinwheel arrangement to form a 60Å diameter disk, while the C-terminal residues (46–70) extend away from the disk, towards the 3-fold axes, in a 5-pointed star

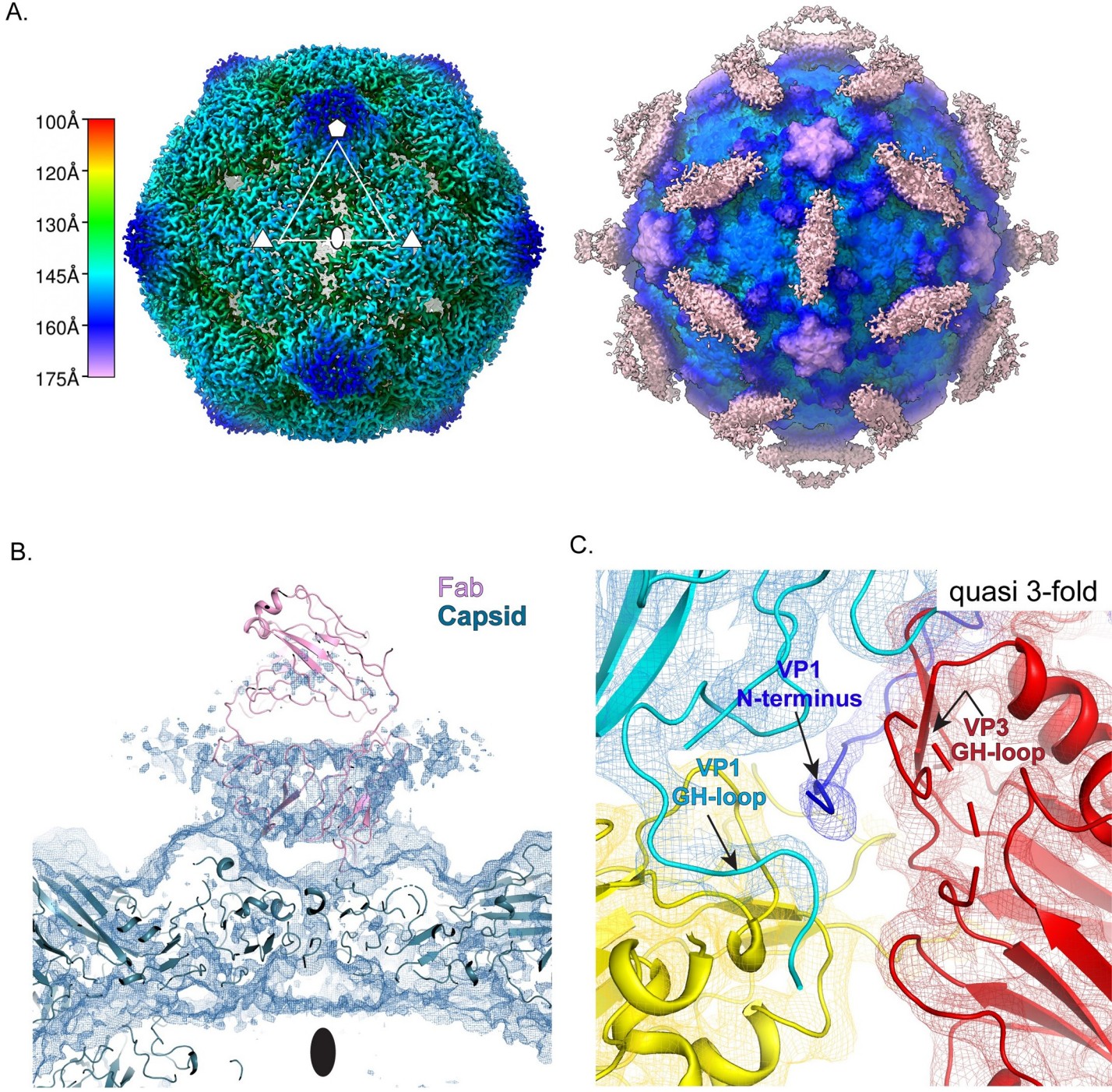

**Fig 5. Receptor catalyzed 135S particle incubated with the anti-VP1 antibody at 37°C (r135+m37).** (A) Radially colored isosurface rendering of the "late" 135S particle that is produced after receptor-catalyzed expanded virus is incubated with anti-VP1 antibody at 37°C for 1h. Bar indicates the radial range that each color corresponds to (A). The panel on the left is contoured at a higher level (σ = 5) to show surface details such as holes at the 2-fold and density in the quasi-3-fold. At a lower contour (σ = 0.1) (right), smeared density due to Fab binding (beige) is apparent. (A and B) The Fab density in r135+m37 appears to be a super-position of the rod-shaped Fab density that extends over the quasi-3-fold (as seen in m135) and the 2-fold-centered density (as seen in the r135+m25 reconstruction). (C) a close-up view of the showing the VP1 N-terminus emerging at quasi-3-fold (compare Figs 2C and 4C).

A.

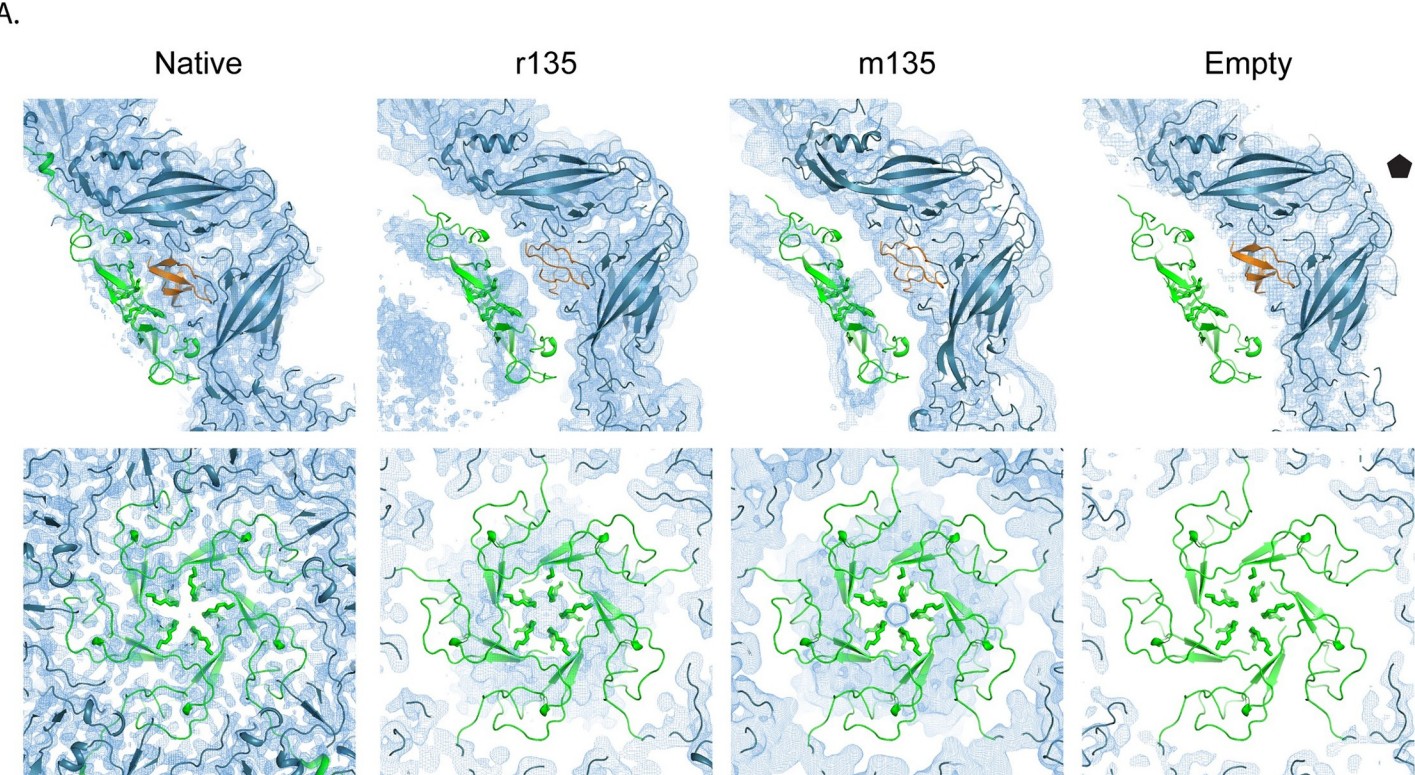

| Native | r135 | m135 | Empty |
|--------|------|------|-------|

**Fig 6. Location of VP4 before and after capsid expansion.** Profile (top panel) and face-on (lower panel) view of VP4 (green) in native (σ = 2), r135 (σ = 1.2), m135 (σ = 0.7) and empty states (σ = 2). The native orientation and location of VP4 has been preserved to highlight the changes in the density, as the capsid expands. Prior to RNA release, VP4 pentamers, somewhat rearranged, appear to remain at their original radius, on the outer surface of the RNA ball, as the rest of the capsid expands outward.

arrangement [2]. In contrast, in 80S empty capsids, most if not all of the RNA is gone and there is no density for VP4 [34,35] (Fig 6, rightmost panel). But what happens to VP4 in between those initial and final states?

Our five high-resolution structures of 135S-like particles offer some intriguing indications that many of the VP4 pentamers may, indeed, remain intact within the virus capsid, with their distances from the protein shell increasing as the rest of the capsid shifts outward. Like many other 135S-like expanded picornavirus structures, our maps show a distinct 10 Å thick layer on the outer surface of the unstructured RNA density (top row center of Fig 1A, and Fig 3A right). Although this layer has commonly been attributed to RNA [20,24], we believe that its shape and dimensions are more consistent with VP4.

In "early 135S particles" (i.e. expanded but with the VP1 N-terminus mostly inside), as seen in the "r135" and "h135" structures, the putative VP4 density is consistently cup-shaped in profile (Fig 6 second panel from left). Seen face-on, the density feature has a radial position and a diameter similar to that of a native VP4 pentamer. Like VP4 pentamers, the density feature is around 10 Å thick (which is much too thin to be duplex RNA). As in native virus, the center of the cup (near the 5-fold axis) contacts the VP3 beta tube; but further from the 5-fold axis, unlike native virus, the density is seen to curve away from the capsid (as if it were peeling off of the inner surface of the capsid). Notably, the shape of the density is pentagonal, with a large central hole, which differs from the star-shaped density of the native pentamer. This would require some concerted conformational change to occur as the pentamer detached.

In the "late 135S particles" (expanded, with the VP1 N-terminus mostly externalized), as seen in the m135, r135+m25 and r135+m37 data sets, the cup-shaped density feature is slightly flatter, and is positioned slightly further away from the inner capsid surface (~14 Å) (Fig 6, third panels from left).

The proposed identification is supported by the findings of Curry et al. [17], which clearly showed low levels of VP4 (up to 25% of the levels observed in virions) in highly purified preparations of poliovirus 135S particles. That study disproved the widely held claim that VP4 (which stains and radiolabels poorly and is difficult to detect in SDS gels optimized for the larger capsid proteins VP1-3) was totally lacking in 135S particles. Moreover, because VP4 plays a key role in viral infection [47], at least some VP4 must remain present in 135S particles to explain the observation that these particles remain infectious [17,18]. Together with the biochemical evidence, the position and shape of the density feature suggests that the thin layer immediately adjacent to the capsid that has previously been attributed to RNA is more likely to be residual VP4 pentamers that are not externalized until the RNA genome is released.

## Discussion

Picornavirus cell-entry is still an area of active research. In the present study we have gained further insight into the dynamics of the expanded genome-containing infectious intermediate of poliovirus, which is known as the 135S particle. Importantly, we show that the 135S particle, which was thought to be a unique structure, is instead an ensemble of structures, that expansion is not completely coupled to externalization the N-terminal extension of VP1, as previously thought, and that a significant fraction of VP4 remains in partially occupied but well-ordered structures on the inside of the particle approximately 14 Å from the inner surface of the remainder of the capsid.

### Decoupling expansion and externalization of the VP1 N-terminus

One of the unexpected findings of the current study was the observation of 135S particles that were expanded but had not externalized the VP1 N-terminus. Thus in the r135 and h135 structures, where the 135S particles had been prepared by protocols that were believed to result in both expansion and externalization of the VP1 N-terminus, icosahedrally averaged maps showed that the VP3 GH loop was in a conformation that blocked the hole at the quasi-3-fold axes and there was no detectable density for the VP1 N-terminus. Expanded particles with internal (but ordered) VP1 N-termini have also been observed as a minority population in preparations of 135S particles of enterovirus D68 [19].

This structural observation appears to contrast with previous structural and biochemical studies that showed exposed VP1 N-termini in 135S-like particles that were prepared using protocols similar to the ones that we used to prepare h135 and r135 particles. Specifically, the exposed N-termini were accessible to proteases and antibodies in the biochemical studies, and at least partially ordered in the structural studies of 135S particles of poliovirus and other enteroviruses. Indeed, we observed weak but significant density for the externalized VP1 N-terminus in a recent intermediate resolution (5.5Å) structure of 135S particles that were prepared using the same heating protocol [14]. To address this apparent discrepancy, we tried applying our current cryoEM software protocols for classification and data processing to the older lower-resolution data set and produced a fenestrated reconstruction that lacked significant density for the VP1 N-terminus. We conclude that the population of particles included a mixture of structures, and that the methods used to select particles to be included in the earlier lower-resolution reconstruction resulted in enrichment in a minority population of particles with the VP1 N-terminus exposed.

## Where does the VP1 N-terminus exit the virion?

Previous structural and biochemical characterization of the 135S particle of poliovirus (and other related picornaviruses) using cryoEM, crystallography, and antibody studies, led to a model wherein, upon expansion, the N-terminus of VP1 initially exits via the 2-fold hole, and later gets locked into the quasi-3-fold channel through a "gear-shift" motion[14,22,33,36]. However, in none of our recent panel of structures did we observe evidence for such a motion. Instead, the density for the N-terminus of VP1 in 135S particles, whenever it was externalized and trapped by an antibody, was consistently observed to occupy the quasi-3-fold channel. Although we cannot rule out a model in which VP1 transiently exits via a hole at the 2-fold axis during the dynamic process of externalization, our recent maps suggest that any such externalized polypeptide is either quickly re-internalized, or quickly migrates to the quasi-3-fold axis. It is also worth noting that even after a long incubation with the monoclonal antibody (m135, r135+m25, and r135+m37), in all cases densities for the VP1 N-terminal extension and for the antibody were consistently weaker than capsid, which suggests that some VP1 N-termini may remain inside the capsid.

## A revised model for the virus-to-135S transition

Taken together our current results suggest a process with three major steps. First, upon binding to the receptor at 37˚C or exposure to high temperature (50˚C) the virus expands and a significant fraction of VP4 is externalized. This step is concomitant with the loss of "pocket factor" and of internal and external stabilizing interactions among neighboring protomers through their N- and C-termini and loops on the outer surface. In a second step, the residues in the VP1 N-terminus are then reorganized such that residues 69–64 run up through the quasi-3-fold. It is clear that this second step must happen during infection because biochemical studies show that the VP1 N-terminus is inserted into membranes in *in vitro* models [11,28] and during infection [48], and plays a key role in the late stages of viral entry [48]. We would argue, however, that the externalization of the VP1 N-terminus is transient and reversible on the time scales relevant to infection (minutes) and that exposure of a significant fraction of the VP1 N-termini on this time scale requires the presence of a trap. Trapping agents could include cellular membranes during infection [47], liposomes in *in vitro* model systems [11,28,29], or antibodies that bind the VP1 N-terminus once exposed [31,33,36]. In order to explain the observation of stable structures in which the VP1 N-terminus is clearly externalized, we postulate a third step in which the VP1 N-terminus slowly becomes locked on the outside of the capsid even in the absence of trapping agents, perhaps by forming stable interactions with the GH loops of VP3 and VP1, as seen in the m135 structure, in our previous intermediate-resolution reconstruction of the poliovirus 135S particle [14], and in the structures of the 135S particles of other enteroviruses [19,20,22–26]. What proportion of the N-termini are exposed in 135S particles, during natural infections, in the presence of cellular membranes, is now an open question.

## Whither the VP1 N-terminus after leaving the capsid?

Although it is clear that residues 69–64 of the VP1 N-terminus exit the capsid at the quasi-3-fold, the course of the externalized residues 1–61 that are distal to the exit point is less clear. In all of our structures, residues 61–56 are disordered. Based on where Fab density is seen, residues 55–42 (which constitute the antibody binding site) can be located either above a 2-fold axis (r135+m25), or above a quasi-3-fold axis (m135), or both (r135+m37). Finally, the results from an earlier low-resolution structure of the 135S particles [40] and the clear density amphipathic helix of VP1 in the r135+m25 structure reported here demonstrates that a significant

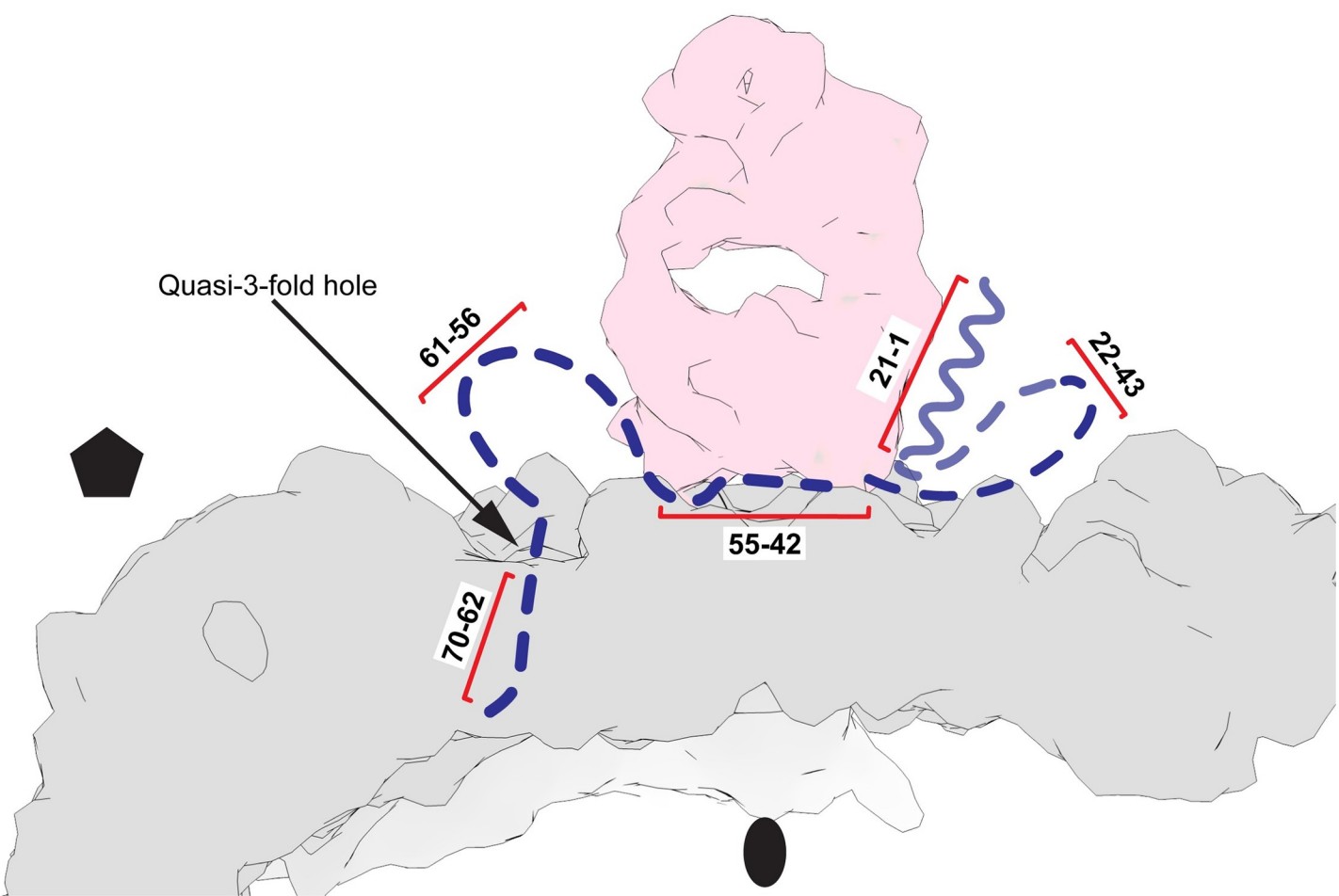

**Fig 7. Schematic of VP1 N-terminal stabilization in the r135+m25 state.** Four distinct areas in the N-terminal extension of VP1 (dashed blue curve) can be stabilized by binding to the Fab (pink) and the viral capsid (grey). This overall binding pattern is suggested by (i) the observation of residues 70–62 in the m135 complex; (ii) the expectation that amino acids from the 55–42 range (the epitope for the anti-VP1 Ab) will bind to the underside of the anti-peptide mAb; (iii) the anchoring of the N-terminal helix (around residue 21) to the top surface of VP2; and (iv) the nonspecific stabilization of the amphipathic N-terminal helix along the side of the Fab in a focus map. Although the pathway of the N-terminal extension is unlikely to be unique, the diagram does help us to understand why the increased prevalence of Fab moieties at the 2-fold hole (in r135+m25) is evidence for the increased anchoring of the amphipathic helix (residues 1–21) on the top surface of VP2, presumably stabilized by lower temperature and longer incubation before the antibody is introduced.

fraction of the first 20 amino acids of VP1 can bind at the top of the EF loop of VP2. These results are summarized in Fig 7.

## Significance of the amphipathic helix of VP1

Earlier studies from our group [11,28] demonstrated the externalization of the VP1 N-terminus and described its role in the binding of 135S particles to membranes. Using floatation assays and V8-protease-based cleavage of the 135S particles, the membrane-interacting part of the peptide was localized to the first 30 amino acids in the sequence [11,28]. Further analysis predicted that the first 20 amino acids would adopt a helical conformation [11], with the helix having a hydrophobic and a hydrophilic side (a common membrane-interactive structural motif). Subsequently, single particle cryoEM analysis of V8-cleaved and uncleaved 135S particles [40] suggested that the externalized VP1 N-terminus would sometimes be anchored on top of the VP2 EF-loop. However, the density for the amphipathic helix had never been

visualized directly. In the present study, we were fortunate to be able to visualize clear density for the entire helix, and to confirm that the helix is located in the same location that the V8-cleavage difference density from an earlier study [40] had predicted it to be (Fig 4D, S3B Fig). The fact that the helix is only clearly visible in focused reconstructions from the screening data set suggests that the interactions between the helix and the VP2 may be weak. However, we speculate that, even if the localization of the helix at the tip of the VP2 EF loop is transient at physiological temperature, by increasing the number of copies of the N-termini at a higher elevation on the virus particle, this interaction may drive the association with membrane, may facilitate engulfment of the virus particle at the cell surface and internalization into endosomes [49], and may serve as an asymmetric trigger for the translocation of the genome into the cytosol [29].

## Difference in antibody density under different conditions

Another puzzling aspect of the structures presented here is the observation that the Fab in the complexes produced by incubation with PVR at 37˚C then with antibody at 25˚C binds above the 2-fold hole, whereas the complexes produced during higher-temperature incubations showed Fab density above the quasi-3-fold axes. A likely explanation for our observations is that some of the N-termini of VP1 become externalized via the quasi-3-fold hole during the initial 3-min incubation with receptor at 37˚C, and their N-terminal helices find the binding site on the VP2 EF loop, where they remain trapped once the temperature is lowered to 25˚C. Upon subsequent incubation with antibody at room temperature, this configuration is further stabilized by nonspecific contacts between the antibody, the VP1 N-terminal helix, and the VP2 EF loops. In the maps (Fig 4A and 4B), this binding mode manifests as a well-ordered density that recognizably corresponds to both halves of the Fab portion of the antibody.

In contrast, when the particles are maintained at 37˚C or 39˚C during antibody binding, the higher temperature reduces the stability of the specific interactions of the VP1 N-terminal helix with the VP2 EF loop, the nonspecific interactions of the hinges of the Fab with the VP2 EF loop "shoulders"; and the nonspecific interactions of the VP1 helix with the sides of the Fab. Thus, the 2-fold binding of Fab was reduced at 37˚C, and almost eliminated at 39˚C. Instead, a greater proportion of the Fab moieties were bound closer to the quasi-3-fold hole, where Fab orientations were less well-constrained, and the Fab density was consequently more diffuse. Asymmetric focus maps were calculated for cylindrical volumes near the 2-fold axes to see if any well-ordered Fab binding sites had been smeared out by symmetry-based density averaging; but no such well-ordered sites were seen.

## Presence of VP4 in an expanded particle

VP4 is a myristoylated, pore-forming protein that is implicated in the translocation the RNA genome across membranes[27,45,47]. This protein was thought to be completely lost after conversion to the 135S state [11,12,16]. However, biochemical and infectivity assays indicate that at least 25% of VP4 remains associated with the virus particle [17], and that the 135S particle remains infectious [17,18] (Fig 6B and S2 Fig). While these data by themselves do not establish whether VP4 remains on the interior or if it is otherwise bound to the capsid surface, they do underscore the presence of VP4 in highly purified 135S particles. In several related picornaviruses, reports of expanded virus structures have noted the disappearance of VP4 from the inner surface of the capsid, and have attributed all of the unstructured density inside of the capsid to RNA, including a thin 10-Å-thick layer at the periphery of the icosahedrally-averaged "RNA ball" [20,24]. However, in our panel of expanded genome-containing structures, we can alternatively account for the density under the 5-fold peak using a native atomic model of the

VP4 pentamer (Fig 6). Moreover, an unmistakable gradual curving of the VP4 layer away from the capsid, peeling off of the inner surface, was seen in the r135 and h135 structures (Fig 6) that is remarkably similar in appearance to the density that has been ascribed to RNA in previous studies with other enteroviruses [20,24]. VP4 has the capacity to bind to RNA [50] and perhaps that is the reason it is being held in place at its original radius in native virus while the rest of the capsid expands radially. Nevertheless, it remains unclear how VP4, thus separated from the capsid, possibly still associated with RNA, and with pentamers intact or not, manages to exit the capsid and to embed itself in the membrane. This will be an important avenue to explore in understanding the crucial genome-translocation step in picornavirus infection.

## Materials and methods

### Virus and antibody

Poliovirus type 1 Mahoney strain was purified by infecting Hela S3 suspension cells at an MOI = 10 for 6hrs at 37˚C. Infected cells were spun down, washed with phosphate buffered saline (PBS:137 mM NaCl, 2.7 mM KCl, 1.1 mM $KH_2PO_4$, 6.5 mM $Na_2HPO_4$, 0.7 mM $CaCl_2$, 0.5 mM $MgCl_2$) and frozen at -80˚C. Virus was released from the cells by repeated freeze-thawing of the infected cell pellet in RSB buffer (10mM Tris pH 7.3, 10 mM NaCl, 1.5 mM $MgCl_2$, 1% IGEPAL CA-630) and clarification of the cell debris through centrifugation. The virus was pelleted through ultracentrifugation and layered on a CsCl density gradient. Subsequently, the virus band was collected and dialysed against PBS before being flash frozen in liquid nitrogen and stored at -80˚C.

The anti-VP1 monoclonal antibody (one of two monoclonal antibodies in a pan-Enterovirus ELISA detection kit) was obtained from Quidel (Athens, Ohio).

### Plaque assay

Vero cells were grown in 6-well plates until confluency at 37˚C and 5% $CO_2$. The cells were then infected with serial dilutions of poliovirus. At 36 h post-infection cells were fixed with 10% formaldehyde and stained with 0.1% crystal violet.

### In vitro generation of poliovirus 135S and 135S-like particles

Anti-VP1 antibody triggered 135S-like particles were generated by mixing 18μL of virus suspension (0.4mg/ml) with 3μL of anti-VP1 antibody (3.2mg/ml) and incubated for 1.5 h at 39˚C in a thermal cycler. (Over numerous trials, these conditions of temperature and duration maximized the proportion of mAb-decorated RNA-containing virus particles. Complexes with Fab fragments were also attempted but did not produce detectable levels of decoration). To generate 135S particles, PV suspension was first buffer exchanged into a low salt buffer (20mM Tris-Cl pH 7.5, 2mM $MgCl_2$). The receptor-triggered conversion of PV was induced by mixing 18.6μL of PV (0.5mg/ml) with 0.4μL of PVR/CD155 (6.1mg/ml) on ice before incubating at 37˚C for 3 min. The reaction was quenched on ice. Heat-triggered 135S particles were generated by heating native PV (in the low salt buffer) to 50˚C for 3 min, before quenching the reaction on ice. The r135+m25/37 particles were produced by incubating PV with PVR, as described above, followed by the addition of the anti-VP1 mAb (150x molar excess over virus) and incubation at 25˚C or 37˚C for 1 h.

### Cryo-electron microscopy and data collection

Carbon-backed lacey grids (Ted Pella) were glow discharged in an easiGLO (PELCO, Fresno, California) glow discharge unit by applying a current of 15mA for 30s. To the freshly glow-

discharged grids, 3.0–3.5μL of virus or virus-ligand mixture was applied. The samples were then allowed to adsorb on the grid for 30 s before blotting away the excess buffer using a blot force of 6 and blot time of 6 s in a FEI Vitrobot Mark IV (ThermoFisher). The grids were plunge frozen in a liquid ethane slush and stored in liquid nitrogen until imaging. Data collection parameters are listed in S1 Table.

### Image processing

Image processing of micrographs was performed within Relion 3.0 [51]. Individual movie frames were aligned and dose-weighted with the Relion implementation of motion correction by applying a negative B-factor of -150 [52]. CTF parameter estimation was performed using Gctf v1.06 [53] on dose-weighted micrographs. Particle-picking was performed in crYOLO [54] by training the neural network on an initial subset of ~300 manually picked particles from several micrographs sampling a range of defocus values. Subsequently, the particle picker was deployed on the complete dataset to identify and save particle co-ordinates which were then imported into Relion. The picked particles were extracted, binned by 4 and subjected to a single round of 2D classification and obviously junk classes were discarded prior to 3D classification. The particle stack was refined against a representative class obtained at the end of the 3D classification. After the first round of refinement, the dataset was subjected to CTF refinement [52] by estimating beam tilt, per-particle defocus and astigmatism followed by Bayesian polishing [52] with default parameters. Icosahedral symmetry was imposed in every round of refinement and the reference model was filtered to 60 Å to prevent model bias. The final resolution (0.143 gold standard FSC) of the structures (S1 Table) was obtained through the Postprocessing job type in Relion, where a density-based mask enclosing the capsid was applied to estimate the FSC between two independently processed half datasets. Local resolution estimates were also obtained by running the Relion's implementation of local resolution algorithm.

### Asymmetric focused classes

After assignment of icosahedral orientations to the particle images, a cylindrical mask was prepared using SPIDER [55] with a diameter of 105Å, centered on the viral 2-fold and enclosing two copies of the quasi 3-fold axis. The initial height of 475Å was then limited to a spherical shell by applying minimum cutoff at 154Å. Then, using relion_particle_symmetry_expand a new STAR file containing 60 icosahedrally related orientations for each particle in the stack was generated. This symmetry expanded stack was then subjected to a masked 3D classification routine in which orientational and translational searches were disabled, the T-value was set to 40, and 5 density classes were generated.

### Model building and refinement

A starting model for m135 was assembled from 1HXS and previously published high-resolution structures of the expanded poliovirus structures [34,56] were docked into the density using COOT [57]. Where none of the pre-existing models for the density fit well, the segments were constructed *de novo*.

Models were iteratively built and refined in REFMAC5 and/or phenix_refine and/or phenix_real_space_refine, until no further improvement was possible [58–60]. To avoid model bias, the Fourier transforms of the experimental maps (both amplitudes and phases) were used as refinement standards. The model included a central protomer (VP1, VP2, and VP3) and neighboring proteins related by icosahedral symmetry. Strong restraints were included for stereochemistry, symmetry, and temperature factors.

### Figure preparation

Figures and illustrations were prepared using PyMol (Schrödinger, LLC), Chimera [61], ChimeraX [62] sourced through the SBGrid Consortium [63] and compiled in Adobe Illustrator (Adobe, USA)

## Supporting information

**S1 Fig. Resolution and quality of maps.** (A) Representative, raw micrographs of the datasets used in the analysis, (B) Masked (continuous line) and unmasked (dotted line) Fourier shell correlation plots of m135 (red), r135 (green), h135 (cyan), r135+m25 (blue) and r135+m37 (purple). At a 0.143 cut-off level, the resolution estimates range from 2.8 to 3.6Å. (C) Local resolution estimates in all the maps were calculated using Relion and the distribution of resolution values within in the masked region is plotted. (D) In the best resolved dataset (m135), individual amino acid side chains are easily discriminated.
(TIF)

**S2 Fig. Binding of anti-VP1 mAb to the VP1 N-terminus and infectivity of mAb released poliovirions.** (A) The anti-VP1 antibody specifically recognizes and binds the N-terminus of VP1 only after the N-terminus has been externalized at 39°C. Poliovirus particles were complexed with the antibody at room temperature or at 39°C for 1.5 h and immunoprecipitated with magnetic Protein A coated beads. After thorough washing of the beads to remove unbound material, the samples were examined on SDS-PAGE gels. Lane 1, Ladder; lane 2, PV particles only; lane 3, antibody; lane 4, soluble fraction of antibody plus 160S PV after binding at RT; lane 5, pelleted fraction of antibody plus 160S PV after binding at RT; lane 6, soluble fraction of antibody plus 160S PV after binding at 39°C; lane 7, pelleted fraction of antibody plus 160S PV after binding at 39°C. Bottom panel, silver staining of the gel to enhance the relatively weak VP4 signal (B) Expanded poliovirus particles are infectious. Native (160S) poliovirus particles previously incubated with anti-VP1 antibody at room temperature or at 39°C for 1.5 h were immunoprecipitated with magnetic Protein A coated beads. After thorough washing of the beads to remove unbound material, the beads were freeze-thawed to release poliovirus particles. Released poliovirus was serially diluted and plated on naïve Vero cells. After 36 h of incubation at 37°C, plaques were visualized by staining with crystal violet.
(TIF)

**S3 Fig. Visualising poliovirus VP1 N-terminus.** (A) Asymmetric focused classes calculated for the r135+m25 dataset with percentage population per class. Class3 is depicted in Fig 4B and 4C. (B) Difference density, shown here in stereo, was calculated from previously published low-resolution reconstructions of poliovirus 135S particles, either untreated or treated with V8 protease, which cleaves at residue 31 of VP1. As a guide, we have superimposed an alpha carbon model of r135+m25. The blue alpha helix (putatively residues 1–21 of VP1) was fitted to a focused class of r135+m25 (as shown in Fig 4D and 4E). Observe that the putative VP1 helix model, bound to the top of VP2, is similar in position and orientation to the previously reported difference map feature.
(TIF)

**S1 Table. Data collection parameters and refinement statistics.** *The r135+m25 model, which includes placeholders for the Fab and VP4 densities, was calculated with phenix_refine, using its automatic choice of weighting schemes. Other final atomic models were refined with Refmac5, using stronger stereochemical restraints.
(TIF)

**S2 Table. List of residues that differ from the native structure.** *Differently ordered residues are those residues that significantly differ in position from the native model (1HXS) following a least squares superposition of individual capsid proteins.
(TIF)

## Acknowledgments

We would like to thank Drs. Lawrence Shapiro and Oliver Harrison (Columbia University) for providing us with PVR, Sara McCoy (Quidel) for providing the anti VP1 antibody, Dr. David Bhella (University of Glasgow) and Dr. Simon Jenni (Harvard University) for helping us run the asymmetric focused classification. We would also like to thank members of the SBGrid team for maintaining the computing environment. Dr. Takanori Nakane (MRC-LMB) is acknowledged for sharing the code to plot the resolution distribution in 3D volumes. We would like to acknowledge the Astbury Biostructure Laboratory (ABSL) for Titan Krios cryo-EM data collection which was funded by the University of Leeds and the Wellcome Trust (108466/Z/15/Z) as well as the Harvard Cryo-EM Center for Structural Biology

## Author Contributions

**Conceptualization:** David J. Filman, Mike Strauss, James M. Hogle.

**Data curation:** Pranav N. M. Shah, David J. Filman.

**Formal analysis:** Pranav N. M. Shah, David J. Filman, Krishanthi S. Karunatilaka, James M. Hogle.

**Funding acquisition:** James M. Hogle.

**Investigation:** Pranav N. M. Shah, David J. Filman, Krishanthi S. Karunatilaka.

**Methodology:** Pranav N. M. Shah, David J. Filman, Krishanthi S. Karunatilaka, Emma L. Hesketh, Elisabetta Groppelli.

**Project administration:** James M. Hogle.

**Resources:** Pranav N. M. Shah, James M. Hogle.

**Software:** Pranav N. M. Shah, David J. Filman.

**Supervision:** David J. Filman, Mike Strauss, James M. Hogle.

**Validation:** David J. Filman, James M. Hogle.

**Visualization:** Pranav N. M. Shah, David J. Filman.

**Writing – original draft:** Pranav N. M. Shah, David J. Filman, James M. Hogle.

**Writing – review & editing:** Pranav N. M. Shah, David J. Filman, Elisabetta Groppelli, Mike Strauss, James M. Hogle.

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
