## [Decision Letter · Decision Letter 0]

27 Apr 2020

Dear Dr. Hogle,

Thank you very much for submitting your manuscript "Cryo-EM structures reveal two distinct conformational states in a picornavirus cell entry intermediate." for consideration at PLOS Pathogens. As with all papers reviewed by the journal, your manuscript was reviewed by members of the editorial board and by two independent reviewers (The 3rd reviewer was terminated due to delay response). In light of the reviews (below this email), we would like to invite the resubmission of a significantly-revised version that takes into account the reviewers' comments.

We cannot make any decision about publication until we have seen the revised manuscript and your response to the reviewers' comments. Your revised manuscript is also likely to be sent to reviewers for further evaluation.

Sincerely,

Guangxiang George Luo, M.D./MPH

Section Editor

PLOS Pathogens

Kasturi Haldar

Editor-in-Chief

PLOS Pathogens

orcid.org/0000-0001-5065-158X

Michael Malim

Editor-in-Chief

PLOS Pathogens

orcid.org/0000-0002-7699-2064

Reviewer's Responses to Questions

**Part I - Summary**

Reviewer #1: General comments:

The manuscript “Cryo-EM structures reveal two distinct conformational states in a picornavirus cell entry intermediate” reported a comprehensive structural analysis of poliovirus intermediate state–135S like particle and identified two states that might represent some snapshots of the poliovirus during its entry process. In literature, the 135S like particle has a canonical conformation form that results from virion binding to the cellular receptor or other triggering factors. Here, the author incubated the native poliovirus in 3 min, either with Pvr receptor at 37 ˚C, or alone at 50 ˚C, and identified a distinctive 135S particle, termed as “the early 135S”. In virtue of a mAb specifically targeting the VP1 N-terminus, the author concluded this conformational state in structural angle. The findings may further widen the understanding of picornavirus infectious process. However, some critical concerns need to be fully addressed.

Reviewer #2: The manuscript titled " Cryo-EM structures reveal two distinct conformational states in a picornavirus cell entry intermediate " by Pranav et al constitutes a comprehensive study on high-resolution cryo-EM structures of poliovirus entry intermediates. Poliovirus, an important class of human pathogens, belonging to the genus of Enterovirus within the family of Picornaviridae, can cause poliomyelitis. In addition, polioviruses also serve as models for understanding the basic mechanisms of host-pathogen interactions, virus entry and viral genome release. It is of importance to characterize the structural details including dynamic conformational changes during viral entry process. Although previously published studies have revealed many important structural features from picornavirus uncoating intermediates, a large gap concerning structural transitions from a mature virion through an uncoating intermediate to a genome-released empty particle exists. The authors describe five cryo-EM structures of altered poliovirus particles and identified at least two unique states: the early and late uncoating intermediate particles. Structural comparisons reveal a number of surprising findings, which, of course, expands the understandings of the cell entry process of many enteroviruses. This manuscript is clearly written and the figures are good. Data analysis seems technically valid. It deserves publication in PLoS Pathogen. However, a number of concerns need to be addressed before the formal publication.

 **********

**Part II – Major Issues: Key Experiments Required for Acceptance**

Reviewer #1: 1. Many other studies have reported the “early” state of uncoating intermediates through either receptor inducing or acid treatment, e.g. the novel expanded (E1) EVD68 particles (Liu et al, 2018, PNAS). The author should mention the difference between the early 135S-like PV in this study to E1 particle (in case of EVD68) or some other receptor-induced intermediates? The paper should cite these papers and discuss with them properly.

2. On the other hand, one of important feature of this early state of intermediate is that VP4 remains inside the particle and accounts for a feature that had been previously ascribed to part of the viral RNA. However, this claim seems take a risk in terms of cryo-EM structure at median resolution. There is no direct biochemical evidence supporting the presence of VP4 in the expanded particles. Although the density (10Å thick) is pentagonal and looks thinner rather than duplex RNA, a possibility is with artifact of the reconstruction calculation with icosahedral symmetry imposed. Additionally, the description of visualizing the exposed N-terminal helix of VP1 should be careful in terms of a low local resolution.

3. In the previous study regarding to poliovirus complexed with soluble Pvr, the cryo-EM reconstruction of the complex can obviously observe the density of Pvr. However, no density was observed in r135 in this study. The authors should explain such difference?

4. In all the structures of immune complexes, the densities of antibody parts are very weak, therefore the interaction details are not clear. The weakness was not remedied by sub-particle calculation, which should need more check and re-implemented to the best performance. The other reason may raise from the utility of intact antibodies instead of Fab for the preparation of complex. The authors would argue that they could only obtain significant decoration of particles using the intact monoclonal antibody. But the reason is still unclear. It strongly recommended that Fab generation for sample preparation would improve the resolution of the complex and might visualize some interesting parts, such as the missing VP1 N-terminus.

5. It’s strange that the particle heat incubations were performed at 39 °C or 37 °C without specified reason. Also, three classes of particles seem derived from different incubation conditions. The author should describe the details and the consideration reasons.

6. Figure 1 was missed in the merged PDF file, which may lead to some new comments.

Reviewer #2: 1. The authors claimed that at least part of the VP4 pentamers indeed remain intact in their five 135S-like particles, which conflicts with many previous experimental observations. Theoretically, it makes sense if viral genome remains intact and inside the particle. However, the relative weak densities for the possible VP4 pentamers in their 135S-like particles are not that convincing to conclude this point. It seems to be solid if authors could provide the mass spectrum results from their purified 135S-like particles to demonstrate the existence of VP4.

2. In many cases, the focused reconstruction (local reconstruction as well) have advantages in solving problems, such as structural flexibility, binding occupancy, imperfect symmetry and symmetry mismatch, usually yielding higher resolution with 60-fold particle number in their case. The authors should provide some comments on this point.

 **********

**Part III – Minor Issues: Editorial and Data Presentation Modifications**

Reviewer #1: Minor comments:

1. The radially colored isosurface rendering of the cryo-EM structures needs the color scale bars throughout the whole figures. Further, the sigma level and icosahedral symmetric axes of cryo-EM density maps should be indicated in the figures and the corresponding legends.

2. Page 7, lines 164-165. The resolution unit was typo as “A”. Please check throughout the manuscript.

3. Page 12, lines 263-265. As the author described the structure of r135 and h135 was identical, several related comparison results should be provided in supplementary information.

4. Suppl. Table 1, refinement parameters of r135+m25, especially the Poor rotamers and Disallowed Rama. were high. The model building should be improved.

Reviewer #2: 1. The radius color bar should be provided in the Figure 1a, 3a, 4a and 5a.

2. To make it clear, it would be better to label necessary structural elements, like beta strands in figure 2e and 2f.

3. It’s very rare to bind to the exact symmetrical (two-fold) axes for antibodies/receptors, resulting in the low occupancy due to the steric clashes. Be better to comment this. That might be reason why the densities for Fab fragments are so smear.

4. Line 164-165, 4.2 Å, not 4.2 A; 6.5 Å, not 6.5 A; (Fig. 1b) not (Fig. 1b.)

5. Line 173, 2.8Å resolution, should be “2.8 Å resolution”

6. Line 259, 3.2Å resolution, should be “3.2 Å resolution”

7. Line 660-669, regarding the asymmetrical reconstruction, the authors can refer to the optimized sub-particle reconstruction strategy to have a try (Science, 2018, 360 (6384): 48-58; DOI: 10.1126/science.aaz1439)

8. Line 409-410, As in native virus, the center of the cup (near the 5-fold axis) contacts the VP3 beta tube. It’s not easy for most readers to see clearly in figure 6. It’s better to color VP3 in red to distinguish with VP1.

9. Line 548, “incubation with” should be “Incubation with”.

 **********

PLOS authors have the option to publish the peer review history of their article (what does this mean?). If published, this will include your full peer review and any attached files.

Reviewer #1: No

Reviewer #2: Yes: Xiangxi Wang
---

## [Decision Letter · Decision Letter 1]

21 Aug 2020

Dear Dr. Hogle,

We are pleased to inform you that your manuscript 'Cryo-EM structures reveal two distinct conformational states in a picornavirus cell entry intermediate.' has been provisionally accepted for publication in PLOS Pathogens.

Best regards,

Guangxiang George Luo, M.D./MPH

Section Editor

PLOS Pathogens

Guangxiang Luo

Section Editor

PLOS Pathogens

Kasturi Haldar

Editor-in-Chief

PLOS Pathogens

orcid.org/0000-0001-5065-158X

Michael Malim

Editor-in-Chief

PLOS Pathogens

orcid.org/0000-0002-7699-2064

Reviewer Comments (if any, and for reference):

Reviewer's Responses to Questions

**Part I - Summary**

Reviewer #1: (No Response)

Reviewer #2: Authors have addressed all concerns I raised, I suggest a publication for this manuscript.

**Part II – Major Issues: Key Experiments Required for Acceptance**

Reviewer #1: (No Response)

Reviewer #2: No extra experiments are needed.

**Part III – Minor Issues: Editorial and Data Presentation Modifications**

Reviewer #1: (No Response)

Reviewer #2: (No Response)

PLOS authors have the option to publish the peer review history of their article (what does this mean?). If published, this will include your full peer review and any attached files.

Reviewer #1: **Yes: **Ying Gu

Reviewer #2: **Yes: **Xiangxi Wang

---

## [Editor Report · Acceptance letter]

24 Sep 2020

Dear Prof. Hogle,

We are delighted to inform you that your manuscript, "Cryo-EM structures reveal two distinct conformational states in a picornavirus cell entry intermediate.," has been formally accepted for publication in PLOS Pathogens.

Best regards,

Kasturi Haldar

Editor-in-Chief

PLOS Pathogens

orcid.org/0000-0001-5065-158X

Michael Malim

Editor-in-Chief

PLOS Pathogens

orcid.org/0000-0002-7699-2064